# Topical Drug Delivery in the Treatment of Skin Wounds and Ocular Trauma Using the Platform Wound Device

**DOI:** 10.3390/pharmaceutics15041060

**Published:** 2023-03-25

**Authors:** Elof Eriksson, Gina L Griffith, Kristo Nuutila

**Affiliations:** 1Harvard Medical School, 25 Shattuck St., Boston, MA 02115, USA; 2Walter Reed Army Institute of Research, 503 Robert Grant Ave, Silver Spring, MD 20910, USA; 3United States Army Institute of Surgical Research, 36950 Chambers Pass, Fort Sam Houston, TX 78234, USA

**Keywords:** ocular injury, platform wound device, topical drug delivery, wound healing

## Abstract

Topical treatment of injuries such as skin wounds and ocular trauma is the favored route of administration. Local drug delivery systems can be applied directly to the injured area, and their properties for releasing therapeutics can be tailored. Topical treatment also reduces the risk of adverse systemic effects while providing very high therapeutic concentrations at the target site. This review article highlights the Platform Wound Device (PWD) (Applied Tissue Technologies LLC, Hingham, MA, USA) for topical drug delivery in the treatment of skin wounds and eye injuries. The PWD is a unique, single-component, impermeable, polyurethane dressing that can be applied immediately after injury to provide a protective dressing and a tool for precise topical delivery of drugs such as analgesics and antibiotics. The use of the PWD as a topical drug delivery platform has been extensively validated in the treatment of skin and eye injuries. The purpose of this article is to summarize the findings from these preclinical and clinical studies.

## 1. Introduction

Topical treatment of injuries such as skin wounds and ocular trauma has many advantages. Treatments can be applied directly to the injured area using only a small volume of liquid or hydrogel. Very high therapeutic concentrations can be achieved at the target site without systemic toxicity [1,2], whereas when the drug is delivered systemically (orally or intravenously), it is diluted in the total body volume. As an example, if a 10 mL antibiotic dose is delivered intravenously, it is effectively diluted in 42,000 mL (the total body water volume of a 70 kg man) [3,4]. In oral administration, the drug concentration is even lower before it reaches the systemic circulation, as it goes through first-pass metabolism occurring in the liver or gut [5]. In addition, systemic antibiotic treatment is limited due to poor tissue penetration, especially in burn and traumatic wounds where the microcirculation is deranged [6,7]. Adverse side effects such as nephropathy, neuropathy, and gastrointestinal disturbances are also a concern [8,9]. Topical drug delivery avoids first-pass metabolism, and the drugs are not diluted in body fluids before reaching the target site. Therefore, topical administration of medications directly to the injury site increases the effective concentration at the target site by up to 4000-fold compared to systemic administration. The direction of the concentration gradient of the drug is also favorable; the concentration is highest at the surface of the wound, where it is needed the most. In addition, local drug delivery systems can be applied directly to the wound, regardless of vascular damage, which potentially avoids adverse systemic effects while providing a high concentration at the injured area [10] (Figure 1).

This review article introduces the Platform Wound Device (PWD) (Applied Tissue Technologies LLC, Hingham, MA, USA) for topical drug delivery in the treatment of skin wounds and eye injuries. The PWD is a treatment platform that can be applied immediately after injury to provide a protective dressing and a tool for precise topical delivery of drugs, such as analgesics and antibiotics, formulated in a liquid or hydrogel. The PWD consists of a transparent polyurethane chamber covering the injured area, such as the face, head and/or extremity, or eye. It can be designed to enclose any size wound over any contour of the body. The polyurethane membrane of the device is transparent, flexible, impermeable, and embossed on the injury-facing side with a pattern of small pyramids that encourage even distribution of applied topical medications. It is attached to the injured area with an adhesive rim and is designed to remain in place for up to 7 days. The half-life of common antibiotics in the device is approximately 20 h. The device also has a port that can be utilized for collecting samples, removing exudate, and administering therapeutics or negative pressure. Moreover, the transparent membrane enables evaluation of the injured area without removing the device. Therefore, only little or no maintenance is required once it is positioned on the wound [11] (Figure 2).

The purpose of this article is to summarize the preclinical and clinical studies that have utilized the PWD in the treatment of skin and eye injuries. First, the use of PWD in the treatment of various skin wounds is described, and the second part will focus on ocular injuries.

## 2. The Platform Wound Device in the Treatment of Skin Wounds

Wound infections are a challenging problem and a major reason for delayed wound healing, causing considerable problems for the patient and healthcare system [12,13]. Many chronic wounds become infected and form biofilms, making them even more difficult to treat and heal. Surgical site infections are a significant cause of morbidity and mortality after operations [14,15]. In addition, impaired wound healing may lead to poor healing outcomes, such as excessive wound contraction and scarring, which require further treatments [16,17,18]. Current management of wound infections includes both systemic (oral or intravenous) and topical antimicrobials [19]. The PWD is a treatment platform that has been extensively validated as a tool to deliver drugs locally to manage infections in skin wounds [11]. The following studies demonstrate how it enables the delivery of topical antimicrobials at very high concentrations without systemic toxicity to prevent and clear infections in both preclinical models and in patient care (Figure 2). All the drugs used in the studies described in this article (wound healing and eye) were purchased as commercial, off-the-shelf products and were of US Pharmacopeia (UPS) grade. They were formulated into hydrogels and liquid solutions under sterile conditions (Table 1).

Junker et al. (2015) used the PWD to deliver high concentrations of gentamicin to treat infected porcine full-thickness wounds. The wounds were infected with 10^8^ colony-forming units (CFU) of *Staphylococcus aureus* (*S. aureus*). Three hours later, a single dose of 10 mL of sterile phosphate-buffered saline (PBS) containing 2 mg/mL of gentamicin powder was delivered into the wounds via the PWD. The gentamicin concentration was 1000× minimum inhibitory concentration (MIC), and systemic treatment would have allowed administration of only 2 ug/mL (1 × MIC). The results showed that the gentamicin treatment eradicated *S. aureus* in wound fluid in 1 h, and wound tissue bacterial counts decreased 64% in 6 h. It was also shown that inflammation decreased significantly in the gentamicin-treated wounds. This study concluded that very high concentrations of topical gentamicin were efficient in treating *S. aureus*-infected full-thickness wounds [20].

Tsai et al. (2015) utilized the PWD in the treatment of infected porcine burns. Standardized full-thickness burns were inoculated with *S. aureus* or *Pseudomonas aeruginosa* (*P. aeruginosa*). After surgical debridement, the burns were treated topically (1 dose, 10 mL) with either 1000× MIC gentamicin (2 mg/mL) or 1000× MIC minocycline (1 mg/mL) using the PWD. The antibiotic solutions were prepared by formulating gentamicin and minocycline powder into sterile PBS. After 6 days of topical treatment with gentamicin or minocycline, *S. aureus* counts decreased in wound tissue from 4.2 to 0.31 and 0.72 log CFU/g, respectively. Correspondingly, *P. aeruginosa* counts decreased from 2.5 to 0.0 and 1.5 log CFU/g in tissue, respectively. In wound fluid of both *S. aureus* and *P. aeruginosa*, CFU counts remained at a baseline of 0.0 log CFU/mL for both antibiotic treatments. The findings demonstrated that high therapeutic concentrations of both gentamicin and minocycline in a liquid delivered topically can rapidly reduce bacterial counts in infected full-thickness porcine burns [21].

Daly et al. (2016) investigated the ability of ultrahigh concentrations of topical minocycline (single dose of 10 mL of 0.1 mg/mL or 1 mg/mL minocycline powder in sterile saline) delivered in liquid via the PWD to promptly reduce bacterial contamination and inflammation in excisional porcine wounds infected with *S. aureus*. Additionally, they investigated the capacity of minocycline to reduce inflammation in noninfected porcine wounds, independent of its antimicrobial effects. Their results concluded that topical minocycline significantly reduced bacterial burden and inflammation in the infected wounds rapidly. Furthermore, it was shown that minocycline decreased local inflammation independently of its antimicrobial effect. In addition, importantly, topical minocycline treatment was compared to intravenous minocycline treatment, and the results showed that topical administration reduced bacterial counts in both wound tissue and fluid significantly more effectively than intravenous minocycline [22].

Yang et al. (2018) studied whether the use of topical minocycline (1 mg/mL) and lidocaine (5% cream) in the PWD over a burn wound can penetrate the eschar without side effects and reduce tissue bacterial burden and pain. Full-thickness burns were created, infected with methicillin-resistant *Staphylococcus aureus* (MRSA), and treated topically with single-dose minocycline powder in sterile saline (10 mL) and lidocaine cream (1 mL) for 3 days. Subsequently, the burn eschar was debrided, and the topical treatment was continued until day 7 post burn creation. The results demonstrated that treatment with the topically delivered minocycline and lidocaine in the PWD significantly reduced both tissue bacterial counts and pain, concluding that the drugs in the PWD were able to penetrate the burn eschar [23].

Nuutila et al. (2018) investigated the utility of the PWD as a treatment platform for both prolonged field care and definitive treatment of burn-injured warfighters. The purpose of the preclinical porcine study was to demonstrate that the PWD can be efficiently used to protect and treat battlefield injuries starting at the time of injury and continuing to definitive treatment. Therefore, immediately after burn creation, the burns were covered with PWDs, and 25 mL of sterile saline containing 0.5 mg/mL lidocaine powder and 1 mg/mL minocycline powder was administered for 3 days. On day 3, the burns were debrided and covered again with PWDs, and continuous negative pressure wound therapy was initiated. Silver sulfadiazine, the US Army prolonged field care standard of care (soc) for battlefield burns, was used as the control treatment. The results showed that PWD treatment with negative pressure significantly reduced erythema and edema in the injured tissue and promoted granulation tissue and neocollagen formation. In addition, it was demonstrated that the PWD treatment reduced bacterial counts in the wounds comparably to the control treatment. The study concluded that the PWD immediately served as a temporary skin barrier protecting the injured area and allowing for precise topical delivery of analgesics, antibiotics, and other medications [11].

Grolman et al. (2019) formulated and characterized an agarose hydrogel that contained high concentrations (1000× MIC; 1 mg/mL) of either minocycline or gentamicin powder. The hydrogel was prepared by formulating 50 mg of agarose powder with 10 mL of water to form 0.5% agarose hydrogel. The solution was autoclaved to dissolve the polymer and sterilize the solution. Subsequently, the hydrogel was allowed to cool down at room temperature for 10 min. Then, minocycline or gentamicin powder was mixed into the agarose hydrogel by vortexing for 1 h to make 1 mg/mL minocycline or gentamicin agarose hydrogels. In vitro studies demonstrated that both antibiotics remained stable in the hydrogel for at least 7 days, and both antibiotics demonstrated sustained release over the time of the experiment. Subsequently, the minocycline hydrogel (single dose of 10 mL of the 1 mg/mL agarose hydrogel) was used together with the PWD in a deep partial-thickness porcine burn model, and its effect as a prophylactic treatment in burn injuries was investigated. The results showed that prophylactic treatment with the agarose minocycline hydrogel mitigated burn wound progression and reduced the bacterial counts as efficiently as commonly used silver sulfadiazine cream [24].

Nuutila et al. (2020) validated the use of the PWD with topical antibiotics in an alginate hydrogel for immediate treatment of porcine burn wounds. Alginate hydrogels containing gentamicin (2 mg/mL), minocycline (8 mg/mL), and vancomycin (1 mg/mL) were formulated: Ultrapure alginate was dissolved in purified water at 1% mass overnight and sterile filtered. The resulting solution was frozen overnight and lyophilized over 4 days under vacuum pressure. Subsequently, dry alginate was added to sterile glass vials along with saline to produce a 2.5% weight solution of alginate. These aliquots were vigorously vortex mixed for 16 h at room temperature, and, subsequently, sterile gentamicin, minocycline, and vancomycin solutions were added during mixing. Hydrogel properties in terms of rheology, drug release, and temperature stability were optimized. Subsequently, the efficacy of the hydrogels (single dose of 10 mL hydrogel) together with the PWD was tested in the treatment of *S. aureus*, *P. aeruginosa*, and *Acinetobacter baumannii* (*A. baumannii*) on infected burn wounds. Blank alginate hydrogel, silver sulfadiazine cream, and intravenous antibiotics were used as control treatments. On days 7 or 45, the animals were euthanized, and the burns were excised for histology and quantitative bacteriology. In addition, on postoperative days 1, 3, 5, and 7, blood samples were drawn to measure the drug concentration in the blood. The results showed that covering the burns with the PWD and treating them topically with the antibiotic-containing alginate hydrogel reduced tissue necrosis and the number of bacteria in the injured tissue in comparison to controls. The blood samples demonstrated that no systemic toxicity was observed, although topical concentrations up to 1000 times higher than intravenous concentrations were used [25].

Eriksson et al. (1996) utilized the PWD with antibiotics in liquid in concentrations up to 2500× MIC in the treatment of a 7-year-old, chronic, non-healing abdominal wound by injecting 20 mL of saline containing clindamycin (10 mg/mL) and gentamicin (1 mg/mL) into the PWD. The PWD was replaced every 4 to 7 days, and treatment was continued for 30 days. The wound started healing, but again at 4 weeks, the bacterial cultures were positive for gentamicin and clindamycin-resistant bacteria. Therefore, the antibiotics were changed to vancomycin (1 mg/mL) and amphotericin B (25 ug/mL). After 10 weeks of treatment, the wounds were completely healed. This clinical study concluded that the PWD allowed for topical delivery of antibiotics in very high concentrations without exceeding the standard dose of intravenous antibiotics [26].

Vranckx et al. (2002) used the PWD in the treatment of 28 infected wounds in 20 patients. Most of the patients had not responded to conventional treatment. The wounds were enclosed with PWDs and treated with different antibiotics, including amphotericin B, cephalexin, ceftazidime, gentamicin, vancomycin, streptomycin, or penicillin, depending on wound culture results. Most of the wounds were chronic wounds, such as leg ulcers. Two of the wounds contained exposed, infected orthopedic hardware (one hip and one knee prosthesis). Both of these wounds healed and stayed healed without removal of the prostheses. The healing rate was 89%, and only three wounds failed to heal. Concentrations of up to 7000× MIC were applied into the PWDs. The study concluded that the PWD was a safe and efficient treatment option when conventional therapy was not working [27].

Cooley et al. (2022) evaluated the PWD in the treatment of various infected wounds, such as pressure sores and foot ulcers. The purpose of the prospective, randomized, multicenter, controlled clinical trial was to investigate the safety, efficacy, and feasibility of the PWD with a single dose of topical commercially available 1% gentamicin cream in a clinical setting and compare it to soc treatments of infected wounds. The wounds were assessed for infection clinically, based on physical exam findings, and with a qualitative scoring system, based on wound swab cultures before and after treatment. Follow-up ranged from 48 to 96 h depending on clinical status. The results showed that delivery of topical gentamicin cream via the PWD was safe and effective at reducing bacterial load in the wounds [28].

## 3. The Platform Wound Device in the Treatment of Ocular Trauma

Each year, more than 2 million eye injuries and infections occur in the United States (US). While most of these preventable injuries heal without intervention, as many as 10–20% of the 2000 work-related ocular injuries occurring daily in the US will result in reduced or complete vision loss [33]. For those with more severe ocular injuries and infections, current medical treatments are generally limited to topical anesthetic drops, antibiotic drops, artificial tears, ocular ointments, punctual plugs, bandage contact lenses, and various methods to keep the eye closed, such as taping, patches, and sutures [29,34]. Most of these treatments, however, are insufficient for severe ocular injuries and infections, have modest effects on wound healing, and, in some cases, may have deleterious effects on the eye. As a result of these medically insufficient treatments, the result for many is impaired or total vision loss.

Treatment modalities for ocular injuries and infections vary greatly depending on the cause and nature of the injury or infection. In cases of ocular infection, immediate administration of topical, fortified antibiotic drops is imperative to treat the infection and prevent reduced or complete vision loss. Oral and intravenous antibiotics are not effective in treating ocular infections. As such, antibiotic drops must be delivered topically to the eye as frequently as every 1 to 2 h, which can be impractical and result in a high burden of care. For patients with compromised eyelid tissue due to burn injuries or other trauma, the treatment options are even more limited and fail to meet the needs of these patients, resulting in poor outcomes. While there are devices available to support the topical delivery of medications in the case of severe corneal injuries, such as scleral prosthetic replacement of ocular surface ecosystem (PROSE) lenses and bandage lenses, such as Prokera^®^, these devices cannot be utilized in patients with severe eyelid trauma or in patients where the eyelid tissue is insufficient to hold the device in place [34]. Other available treatment options, such as amniotic membrane placement, are impractical due to the high rate of degradation of the membrane, while tarsorrhaphy (partial suturing of the upper and lower eyelids) to protect the eye might not be an option due to the lack of eyelid tissue. Alternatively, clinicians have utilized varying treatment modalities, including eye patches, modified swim goggles, or cellophane wrap, in conjunction with topical ophthalmic treatments to create moisture chambers to protect the ocular surface, but the results are often suboptimal, as the described devices fail to successfully deliver therapeutics to the ocular surface [30]. Devices that allow for the constant delivery of therapeutics in specific volumes and concentrations while not requiring the presence of intact periocular structures and eyelids would provide a marked benefit to burn patients and those without the ability to administer eye drops. To meet this unmet medical need, devices such as the ocular wound chamber (OWC), which is based on platform wound device (PWD) technology, have been explored for use in these patients.

PWD technology previously developed for skin wounds has been adapted for ocular use as a topical delivery device. PWD technology for ocular use can be utilized to treat patients with ocular injuries, ocular infections, and in patients with facial burns while providing ocular protection and promoting healing of the ocular surface. The ocular wound chamber (OWC) is a flexible semi-transparent device that attaches to the perimeter of the wound. Prototypes for human and animal use have been developed (Figure 3). OWCs developed for animal use were utilized in a guinea pig model for a variety of ocular studies. Initial studies utilizing the OWC in a guinea pig model showed not only that use of the OWC was safe for the ocular surface and surrounding tissues, but it also allowed for the delivery of specific volumes and concentrations of therapeutics to the surface of the eye and surrounding periocular tissues through the creation of a watertight seal [30]. Subsequent studies investigated the use of the OWC when filled with potential therapeutics and found that therapeutics utilized in the OWC were safe and could potentially modulate the healing of ocular wounds [29,31]. The use of the OWC on more severe infections and injuries revealed that OWC use could prevent the formation of exposure keratopathy in a guinea pig model [30]. Even more importantly, studies investigating the safety and effectiveness of the ocular wound chamber on ocular infections found that OWC use with 0.5% moxifloxacin hydrochloride drops significantly decreased the overall bacterial load as early as 24 h after treatment began [32]. Taken together, the results of these studies indicate that OWC use provides advantages over the currently available therapies utilized to treat ocular and periocular injuries and ocular infections. These studies support the use of the OWC as an effective delivery device to provide constant delivery of topical therapeutics to the surface of the eye and damaged periocular tissues to reduce the burden of care and reduce vision loss (Table 1).

## 4. Discussion

Topical administration is a convenient route for delivering therapeutic agents. Developments in tissue engineering have introduced multiple biomaterials that can be used as vehicles for local drug delivery. Drug release and other properties of these materials can be tailored as needed for treatment. Current options include solid powders, semisolid ointments, creams, gels, liquid lotions, and suspensions that can be loaded with various drugs [35,36]. There are many commercially available topical treatments with antibacterial properties, such as gentamicin cream and silver sulfadiazine ointment, that are commonly used in clinical practice to treat wounds. Different dressings, such as films, have been used as secondary dressings to cover the injury after the application of topical therapeutics [36]. However, none of these are optimal, and they are not designed to act as delivery platforms.

Dermal patches and microneedle arrays have been designed and used for topical drug delivery. The dermal patch is a well-known approach for delivering therapeutics through the skin without needles. They are usually bilayered and incorporated with therapeutics, such as antimicrobials and anti-inflammatory agents. Dermal patches are more commonly used for different skin conditions than in the treatment of wounds [37]. Microneedles are arrays of short needles that were initially developed for painless delivery of therapeutics transdermally. The needles pass through the skin barrier and can be loaded with therapeutics, such as insulin, vaccines, and pain medications. Traditionally, wound-healing microneedles have not been a common choice [15].

The PWD is a unique device that was designed for topical drug delivery. It can be used as a platform to apply any biomaterial containing drugs. It encloses the injured area, creating a controlled environment for therapeutics. In comparison to transdermal patches and microneedles, the PWD has more versatility. Unlike dermal patches and microneedles, it can be used on infected joint prostheses, infected hardware in the back, fistulas, and in the treatment of eye injuries. The device allows for precise delivery of pharmaceuticals in a sustained release mode and can be left on for several days with full effect.

The polyurethane membrane of the PWD is impermeable, so it keeps the injured area moist, which is beneficial, especially in skin wound healing [38]. The adhesive rim of the PWD is strong and made of medical-grade acrylic, keeping the device in place for many days. The combined permeability of the adhesive and the backing of the PWD has been designed to be greater than that of intact skin, so moisture will not lift it off. In addition to delivering drugs topically for skin wounds and ocular injuries, the PWD has been used as a tool for gene transfer by creating a fluid-filled environment for gene delivery to the wound floor by in vivo microseeding [39,40,41,42]. It has also been utilized as a platform for minced skin grafting and cell therapy. Minced skin particles have been transplanted to wounds, and the PWD has been used to enclose the wound and secure the grafts while also, importantly, creating a wet or moist wound environment that makes the orientation of the transplanted skin grafts inconsequential [43,44,45]. Another feature of the PWD is that it becomes a negative pressure wound (NPWT) device without foam or gauze once connected to a negative pressure pump. Preclinical and clinical studies have shown that PWD technology is comparable to the traditional foam and gauze NPWT devices [11,46,47,48].

Furthermore, the utility of the PWD for both prolonged field care and definitive treatment of burn- and blast-injured warfighters has been explored by the US Department of Defense (DOD). As PWD technology is multifunctional, lightweight, compact, and easy-to-use, it could be used on the battlefield by medics as a protective dressing and a platform for delivering antimicrobials and analgesics. Later, it could be utilized as a negative-pressure wound therapy device and as a delivery platform for regenerative medicine approaches [11,23,25]. In addition, importantly, since PWD technology encloses each wound, it can reduce the risk of nosocomial-acquired infections. Approximately 1 in 10 hospitalized patients acquires an infection after admission, which accounts for $6.7 billion in excess costs in the US each year [49]. Preventing nosocomial infections could reduce the lengths of stay and the need for additional diagnostic and therapeutic interventions for approximately 1.7 million patients annually [50]. Preclinical infection animal models have shown that use of the PWD might prevent nosocomial infections by preventing bacterial transfer between wounds [22,23,24,25].

## 5. Conclusions

Applying medications locally and directly to the injured area is convenient. It allows the use of high therapeutic concentrations without systemic toxicity. Various dressings have been used to cover the injured area after administration of therapeutics, but they have not been designed to act as delivery platforms for medications. The purpose of this review is to introduce the Platform Wound Device (PWD). The PWD is a topical drug delivery platform that can be used to deliver any topical medication, including antimicrobials, analgesics, growth factors, enzymatic debridements, and scar treatments for all types of skin and eye injuries. It is currently approved by the FDA for non-exudating to minimally exuding wounds, pressure sores, lacerations/abrasions, partial- and full-thickness wounds, surgical incisions, second-degree burns, donor sites, IV sites, autologous skin graft transplants, and on the orbital rim to facilitate treating exposure keratopathy and ocular wounds from facial burns. In addition, it has been cleared as an NPWT dressing due to its proprietary pump. The device is made of polyurethane and has an embossed superstructure that facilitates even distribution of topical therapeutics. The PWD is manufactured to GMP and validated by FDA guidelines for biocompatibility, and the shelf life of the device has been certified for 3 years after sterilization. It can be designed to enclose any kind of injury, from small chronic wounds to large burns and ocular injuries. It can even be used to cover an entire extremity. Currently, it is tooled to be manufactured in sizes of 2” round, 3” round, 1” × 3” oblong, 3” × 5” oblong, 2”round ocular wound chamber, leg limb device, and arm limb device. Multiple preclinical and clinical studies have demonstrated its safety and efficacy as a protective dressing and delivery platform for topical therapeutics in the treatment of skin wounds and ocular trauma.

## Figures and Tables

**Figure 1 pharmaceutics-15-01060-f001:**
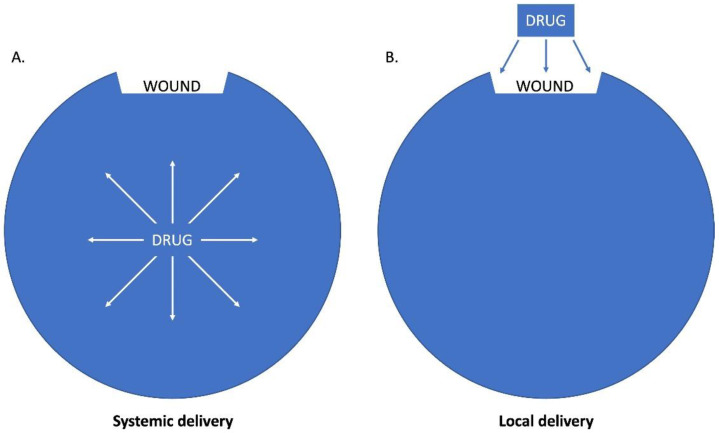
(**A**) Systemic drug delivery. The image depicts how the drug is diluted in the total body volume when delivered systemically (intravenously or orally). (**B**) Topical drug delivery. The image depicts how direct topical administration of the drug increases the effective concentration at the target site compared to systemic administration. The concentration is the highest at the surface of the wound, where it is needed the most.

**Figure 2 pharmaceutics-15-01060-f002:**
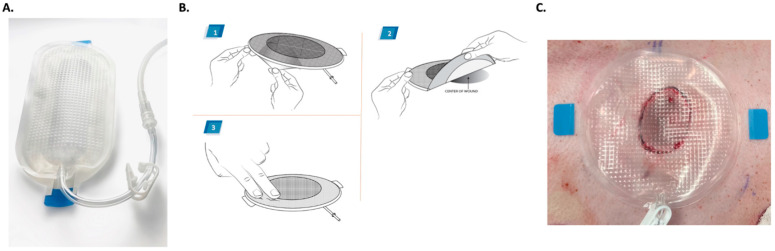
(**A**) The image depicts an oval-shaped PWD. The PWD consists of a transparent polyurethane chamber covering the injured area. The polyurethane membrane of the device is flexible, impermeable, and embossed on the skin-facing side with a pattern of small pyramids that promote even distribution of liquid- or hydrogel-formulated medications. It is designed to remain in place for up to 4 to 7 days. Wound fluid may be removed, and medications may be added via the port. The transparent cover allows evaluation of the injured area without removing the PWD. (**B**) This image describes the ease of application: (1) Remove the backing paper, (2) Place the PWD in the center of the injured area, (3) Seal it to the skin just outside the perimeter of the wound. The PWD has an adhesive flexible ring that makes application fast and convenient on the injured area. It can be designed to enclose any size wound over any contour of the body. (**C**) The PWD placed on a porcine full-thickness wound.

**Figure 3 pharmaceutics-15-01060-f003:**
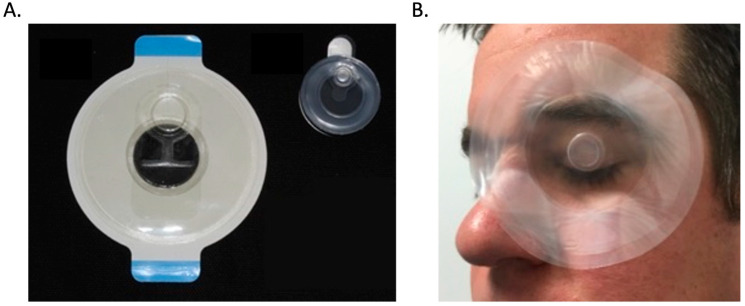
(**A**) A 3-inch diameter ocular wound chamber designed for human use with a port for drug delivery and a 1-inch diameter ocular wound chamber for animal use. (**B**) The PWD shown protecting an injured eye and providing a watertight seal for the delivery of therapeutics to the ocular surface.

**Table 1 pharmaceutics-15-01060-t001:** Summary of the skin wound and ocular injury studies that have utilized the PWD in topical delivery.

	Study	Species	Type of Injury	Pathogen	Treatment
Skin Wounds	Junker et al. (2015) [20]	Pig	Full-thickness wound	*S. aureus*	Single dose of gentamicin powder [2 mg/mL] in saline
	Tsai et al. (2016) [21]	Pig	Full-thickness burn	*S. aureus;* *P. aeruginosa*	Single dose of gentamicin powder [2 mg/mL] or minocycline powder [1 mg/mL] in saline
	Daly et al. (2016) [22]	Pig	Full-thickness wound	*S. aureus*	Single dose of minocycline powder [0.1 mg/mL; 1 mg/mL] in saline
	Yang et al. (2018) [23]	Pig	Full-thickness burn	MRSA	Single dose of minocycline powder [1 mg/mL] in saline and 5% lidocaine cream
	Grolman et al. (2019) [24]	Pig	Deep partial-thickness burn	-	Single dose of minocycline powder [1 mg/mL] in agarose hydrogel
	Nuutila et al. (2018) [11]	Pig	Full-thickness burn	-	Single dose of minocycline powder [1 mg/mL] and lidocaine powder [0.5 mg/mL] in saline
	Nuutila et al. (2020) [25]	Pig	Deep partial-thickness burn	*S. aureus;* *P. aeruginosa;* *A. baumannii*	Single dose of gentamicin powder [2 mg/mL] or minocycline powder [8 mg/mL] or vancomycin powder [1 mg/mL] in alginate hydrogel
	Eriksson et al. (1996) [26]	Human	Abdominal wound	Multiple pathogens	Multiple doses of gentamicin [1 mg/mL], clindamycin [1 mg/mL], vancomycin [1 mg/mL], and amphotericin B (25 ug/mL) in saline
	Vranckx et al. (2002) [27]	Human	Infected skin wounds	Multiple pathogens	Multiple doses of high concentrations of various antibiotics in saline (such as amphotericin B, cephalexin, ceftazidime, pentamicin, vancomycin, streptomycin, and penicillin)
	Cooley et al. (2022) [28]	Human	Infected skin wounds	Multiple pathogens	Single dose of gentamicin cream [0.1%]
Ocular trauma	McDaniel et al. (2018) [29]	Guinea pig	Corneal epithelial wound	-	Single dose of hydroxypropyl methylcellulose gel or liquid
	Holt et al. (2018) [30]	Guinea pig	Keratopathy	-	Single dose of hydroxypropyl methylcellulose gel or balanced salt solution
	Griffith et al. (2021) [31]	Guinea pig	Corneal epithelial wound	-	Single dose of human platelet lysate [20%; 100%] in balanced salt solution
	McDaniel et al. (2020) [32]	Guinea pig	Keratopathy	*P. aeruginosa*	Single dose of moxifloxacin hydrochloride drops [0.5%]

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
