# Peer review of "Topical Drug Delivery in the Treatment of Skin Wounds and Ocular Trauma Using the Platform Wound Device"

_pharmaceutics, 2023, doi:10.3390/pharmaceutics15041060_

Round 1

Reviewer 1 Report

Authors summarized the preclinical and clinical studies that used the platform wound device. I have the following minor comments:

l. Discussion below figure 1 seems to be repetitive as that of introduction. Please check.

2. Please check on the reference numbers and years of publication in table 1, page -3.

3. Please add discussion on the maintenance of pwd device, for example temperature and pH conditions for the pwd device in terms of wound healing. 

Author Response

Authors summarized the preclinical and clinical studies that used the platform wound device. I have the following minor comments:

Authors’ response: Thank you very much for reviewing our manuscript.

  1. Discussion below figure 1 seems to be repetitive as that of introduction. Please check.

Authors’ response: Thank you. The Figure 1 legend has been revised. Please see the revised manuscript.

  1. Please check on the reference numbers and years of publication in table 1, page -3.

Authors’ response: Thank you. The Table 1 has been revised. Please see the revised manuscript.

  1. Please add discussion on the maintenance of pwd device, for example temperature and pH conditions for the pwd device in terms of wound healing.

Authors’ response: Thank you. We have now added discussion about this to the discussion and conclusions of the revised manuscript.

The shelf-life of PWD has currently been certified for three years after sterilization. The device has been validated by FDA guidelines for biocompatibility and our studies have shown that it has good tolerance for variety of temperatures and pH conditions. The adhesive rim of the PWD is  strong and made of medical grade acrylic, keeping the device in place for many days. The combined permeability of the adhesive and the backing of the PWD has been designed to be greater than that of intact skin so moisture won’t lift it off. The PWD is manufactured to GMP and is compliant with ISO 13485:2016. All aspects of the PWD have been validated through material testing.

Reviewer 2 Report

Comments and Suggestions for Authors

In the present manuscript, the authors reported the interesting study focusing on topical treatment of skin and ocular injuries utilizing PWD.

The manuscript can be improved, it is not formatted correctly according to the journal instructions. E.g., line numbers are missing. The idea of the manuscript is brilliant however the discussion and conclusion of the manuscript needs to be improved

The authors should pay more attention on the following points that can be addressed and improve the manuscript.

1.       Please be more specific when you say "whereas when the drug is delivered systemically," such as when the drug is administered orally or intravenously.

2.       Please specify if PWD can be used with any topical medication or there are limits.

·         It is not clear if the PWD is suitable for all types of skin injuries. Here a few types of skin injuries are highlighted below. Cuts, lacerations, gashes and tears. These are wounds that go through the skin to the fat tissue. Caused by a sharp object.

·         Scrapes, abrasions, scratches and floor burns. These are surface wounds that don't go all the way through the skin.

·         Bruises. These are bleeding into the skin from damaged blood vessels.

3.       Can the PWD deliver the medication to the injured or infected dermis and hypordemis?

4.       Can stitched wounds be treated with the PWD?

5.       How is the PWD different transdermal patches, is it better?

6.       How is this discussion different from the introduction? Compare the PWD to the other different kinds of topical treatment, patches and microneedles patches.

7.       The publications cited in this manuscript reported interesting findings, but it is not clear if the PWD wound healing method is better the than the conventional methods.

8.       Expand on conclusion. 

Author Response

Reviewer 2

Comments and Suggestions for Authors

In the present manuscript, the authors reported the interesting study focusing on topical treatment of skin and ocular injuries utilizing PWD. 

The manuscript can be improved, it is not formatted correctly according to the journal instructions. E.g., line numbers are missing. The idea of the manuscript is brilliant however the discussion and conclusion of the manuscript needs to be improved

Authors’ response: Thank you. The manuscript should now be in the correct format including line numbers.

The authors should pay more attention on the following points that can be addressed and improve the manuscript.

  1. Please be more specific when you say "whereas when the drug is delivered systemically," such as when the drug is administered orally or intravenously.

Authors’ response: Thank you. This has now been specified throughout the manuscript. Please the revised manuscript.

  1. Please specify if PWD can be used with any topical medication or there are limits.

It is not clear if the PWD is suitable for all types of skin injuries. Here a few types of skin injuries are highlighted below. Cuts, lacerations, gashes and tears. These are wounds that go through the skin to the fat tissue. Caused by a sharp object.

Scrapes, abrasions, scratches and floor burns. These are surface wounds that don't go all the way through the skin. 

Bruises. These are bleeding into the skin from damaged blood vessels.

Authors’ response: Thank you. The PWD is topical drug delivery platform that can be used to deliver any topical medication, such as antimicrobials, analgesics, growth factors, enzymatic debridement, scar treatments for all types of skin injuries. It has currently been approved by the FDA for, Non-exudating to minimally exuding wounds, Pressure sores, Lacerations/ abrasions, Partial and full thickness wounds, Surgical incisions, Second degree burns, Donor sites, IV sites, Autologous skin graft transplants, On orbital rim to facilitate treating exposure keratopathy and ocular wounds from facial burns. In addition, it has been cleared as a NPWT dressing with its proprietary pump. The PWD can be designed to enclose any size of injury. Currently, it is tooled to be manufactured in sizes of 2” round, 3” round, 1” x 3” oblong, 3” x 5” oblong, 2” round ocular wound chamber, leg limb device and arm limb device. Please see the revised manuscript.

  1. Can the PWD deliver the medication to the injured or infected dermis and hypordemis?

Authors’ response: Thank you. Yes. In preclinical studies we have used it to cover full-thickness excisional wounds that go down to the fat layer.

  1. Can stitched wounds be treated with the PWD?

Authors’ response: Thank you. Yes. In preclinical studies we have used it to cover excisional closed incisions.

  1. How is the PWD different transdermal patches, is it better? 

Authors’ response: Thank you.  It has more versatility. It  can be used on infected joint prosthesis, infected hardware in the back, fistulas and in the treatment of eye. Transdermal patches cannot. The device allows for precise delivery of pharmaceuticals in a sustained release mode and can be left on for several days with full effect. One single dose does not exceed one IV dose and up to 1000x MIC can be achieved without toxicity. Please see discussion in the revised manuscript.

  1. How is this discussion different from the introduction? Compare the PWD to the other different kinds of topical treatment, patches and microneedles patches. 

Authors’ response: Thank you. The discussion has now been revised. Please see the discussion in the revised manuscript.

  1. The publications cited in this manuscript reported interesting findings, but it is not clear if the PWD wound healing method is better the than the conventional methods. 

Authors’ response: Thank you. The discussion has now been revised. Please see the discussion in the revised manuscript.

  1. Expand on conclusion. 

Authors’ response: Thank you. The conclusions has been expanded. Please see the revised manuscript.

Reviewer 3 Report

1. Add Table 1 in which pharmaceutical form is the specific formulation, and how often it is applied. The matrix is missing, which substances are used in the production of these therapeutic systems.

2. You must list pharmaceutical forms

3. Nowhere do you mention the sterility of the preparations, especially preparations for eyes and open wounds

4. Reference 23, I don't see the form for minocycline, and how did they sterilize the cream? We know that creams are extremely rare forms for the eyes, just because of the sterilization process.

5. Please, when you write gel, which type of gel, which polymer carrier... expand a bit. For example, the second paragraph on the fifth page.

6. The main objection is that the work lacks the composition of the formulation. Since sterilization has been achieved, we are talking about active substances only. Please complete.

7. Mesate therapeutics and drugs

8. Add what techniques and procedures are used for production

9. Unfortunately, all of your examples are antibacterial. It seems to me that there are no other antimicrobials, let alone improved wound healing. Please pay attention to it.

10. Please expand the work, it seems to me that everything is just a little noted

Author Response

  1. Add Table 1 in which pharmaceutical form is the specific formulation, and how often it is applied. The matrix is missing, which substances are used in the production of these therapeutic systems.

Authors response:Thank you. Pharmaceutical form and the vehicle material have been added to the table. Pharmaceutical forms, matrixes and the number of applications have been clarified in the text. Please see the revised Table 1 and the revised manuscript.

  1. You must list pharmaceutical forms

Authors’ response: Thank you. Pharmaceutical form and the vehicle material have been added to the table. Pharmaceutical forms, matrixes and the number of applications have been clarified in the text. Please see the revised Table 1 and the revised manuscript.

  1. Nowhere do you mention the sterility of the preparations, especially preparations for eyes and open wounds

Authors’ response: Thank you. All the drugs used in these studies (wound healing and eye) were all commercial off the shelf purchased and UPS grade. They were formulated into hydrogels and liquid solutions under sterile conditions. This has now been clarified in the manuscript text.

As an example, this how the antibiotic containing alginate hydrogels were formulated [11]: FMC Pronova Ultrapure MVG alginate (Dupont, Wilmington, DE) was dissolved in Millipore-purified water (Millipore, Burlington, MA) at 1% mass over- night and sterile-filtered with a 0.22-lm filter. The resulting solution was frozen overnight in a -30°C freezer and lyophilized over 4 days at a vacuum pressure of 0.035 Torr. The 6.625 g of dry alginate was added to sterile glass vials along with TheraTears Solution (Akorn, Lake Forest, IL) to produce a 2.5% weight solution of alginate. These aliquots were vigorously vortex-mixed for 16h at room temperature, and subsequently, sterile gentamicin, minocycline, and vancomycin solutions were added during the vortexing.

  1. Reference 23, I don't see the form for minocycline, and how did they sterilize the cream? We know that creams are extremely rare forms for the eyes, just because of the sterilization process.

Authors’ response: Thank you. In reference 23, sterile UPS grade minocycline powder was formulated into sterile saline and the lidocaine cream used in the study was FDA approved commercially available product. This has now been clarified in the manuscript text.

  1. Please, when you write gel, which type of gel, which polymer carrier... expand a bit. For example, the second paragraph on the fifth page.

Authors’ response: Thank you. This has now been clarified in the manuscript text. Please see the revised manuscript.

  1. The main objection is that the work lacks the composition of the formulation. Since sterilization has been achieved, we are talking about active substances only. Please complete.

Authors’ response: The composition of the different formulations have been added to the manuscript text and to the revised Table 1. Please see the revised manuscript.

  1. Add what techniques and procedures are used for production.

Authors’ response: The PWD is manufactured to GMP and is compliant with ISO 13485:2016. All aspects of the PWD have been validated through material testing and the device has been validated by FDA guidelines for biocompatibility. Please see the revised manuscript.

Also, more information on the manufacturing of the used hydrogels has been added. Please see the revised manuscript.

  1. Unfortunately, all of your examples are antibacterial. It seems to me that there are no other antimicrobials, let alone improved wound healing. Please pay attention to it.

Authors’ response: Thank you. Majority of the studies described in the manuscript have utilized the PWD as a delivery platform for various antimicrobials. However, the PWD is a topical drug delivery platform that can be used to deliver any topical medication, such as antimicrobials, analgesics, growth factors, enzymatic debridement, scar treatments for all types of skin injuries. It has currently been approved by the FDA for, Non-exudating to minimally exuding wounds, Pressure sores, Lacerations/ abrasions, Partial and full thickness wounds, Surgical incisions, Second degree burns, Donor sites, IV sites, Autologous skin graft transplants, On orbital rim to facilitate treating exposure keratopathy and ocular wounds from facial burns. In addition, it has been cleared as a NPWT dressing with its proprietary pump.

  1. Please expand the work, it seems to me that everything is just a little noted

Authors’ response. Thank you. We have now added more details to the studies describing the use of the PWD. In addition, both discussion and conclusions of the manuscript have been expanded. Please see the revised manuscript.

Round 2

Reviewer 3 Report

Thank you for following almost all my suggestions and recommendations. I think you know that we have contributed to improving the quality of this work. I ask you again to pay attention to technical errors, such as units (all units except % separated from the number), and also, be careful of introducing abbreviations.

Thank you very much.

The paper is ready for acceptance.